# Suppression of Bone Necrosis around Tooth Extraction Socket in a MRONJ-like Mouse Model by E-rhBMP-2 Containing Artificial Bone Graft Administration

**DOI:** 10.3390/ijms222312823

**Published:** 2021-11-26

**Authors:** Yukie Tanaka, Kyaw Thu Aung, Mitsuaki Ono, Akihiro Mikai, Anh Tuan Dang, Emilio Satoshi Hara, Ikue Tosa, Kei Ishibashi, Aya Ono-Kimura, Kumiko Nawachi, Takuo Kuboki, Toshitaka Oohashi

**Affiliations:** 1Department of Molecular Biology and Biochemistry, Okayama University Graduate School of Medicine, Dentistry and Pharmaceutical Sciences, Okayama 700-8558, Japan; pblq4l6l@s.okayama-u.ac.jp (Y.T.); ppjj5xuj@s.okayama-u.ac.jp (A.T.D.); p9po6y8u@s.okayama-u.ac.jp (K.I.); oohashi@cc.okayama-u.ac.jp (T.O.); 2Department of Oral Rehabilitation and Regenerative Medicine, Okayama University Graduate School of Medicine, Dentistry and Pharmaceutical Sciences, Okayama 700-8558, Japan; pidh4nf2@s.okayama-u.ac.jp (K.T.A.); de421035@s.okayama-u.ac.jp (I.T.); kuboki@md.okayama-u.ac.jp (T.K.); 3Department of Oral Rehabilitation and Implantology, Okayama University Hospital, Okayama 700-8558, Japan; a.mikai@s.okayama-u.ac.jp (A.M.); a-kimura@md.okayama-u.ac.jp (A.O.-K.); nawachik@md.okayama-u.ac.jp (K.N.); 4Department of Biomaterials, Okayama University Graduate School of Medicine, Dentistry and Pharmaceutical Sciences, Okayama 700-8558, Japan; haraemilio@okayama-u.ac.jp; 5Center for Innovative Clinical Medicine, Okayama University Hospital, Okayama 700-8558, Japan

**Keywords:** medication-related osteonecrosis of the jaw, BMP-2, β-tricalcium phosphate, bone formation, bone necrosis

## Abstract

Medication-related osteonecrosis of the jaw (MRONJ) is related to impaired bone healing conditions in the maxillomandibular bone region as a complication of bisphosphonate intake. Although there are several hypotheses for the onset of MRONJ symptoms, one of the possible causes is the inhibition of bone turnover and blood supply leading to bone necrosis. The optimal treatment strategy for MRONJ has not been established either. BMP-2, a member of the TGF-β superfamily, is well known for regulating bone remodeling and homeostasis prenatally and postnatally. Therefore, the objectives of this study were to evaluate whether cyclophosphamide/zoledronate (CY/ZA) induces necrosis of the bone surrounding the tooth extraction socket, and to examine the therapeutic potential of BMP-2 in combination with the hard osteoinductive biomaterial, β-tricalcium phosphate (β-TCP), in the prevention and treatment of alveolar bone loss around the tooth extraction socket in MRONJ-like mice models. First, CY/ZA was intraperitoneally administered for three weeks, and alveolar bone necrosis was evaluated before and after tooth extraction. Next, the effect of BMP-2/β-TCP was investigated in both MRONJ-like prevention and treatment models. In the prevention model, CY/ZA was continuously administered for four weeks after BMP-2/β-TCP transplantation. In the treatment model, CY/ZA administration was suspended after transplantation of BMP-2/β-TCP. The results showed that CY/ZA induced a significant decrease in the number of empty lacunae, a sign of bone necrosis, in the alveolar bone around the tooth extraction socket after tooth extraction. Histological analysis showed a significant decrease in the necrotic alveolar bone around tooth extraction sockets in the BMP-2/β-TCP transplantation group compared to the non-transplanted control group in both MRONJ-like prevention and treatment models. However, bone mineral density, determined by micro-CT analysis, was significantly higher in the BMP-2/β-TCP transplanted group than in the control group in the prevention model only. These results clarified that alveolar bone necrosis around tooth extraction sockets can be induced after surgical intervention under CY/ZA administration. In addition, transplantation of BMP-2/β-TCP reduced the necrotic alveolar bone around the tooth extraction socket. Therefore, a combination of BMP-2/β-TCP could be an alternative approach for both prevention and treatment of MRONJ-like symptoms.

## 1. Introduction

Medication-related osteonecrosis of the jaw (MRONJ) is one of the severe complications in patients taking bisphosphonates, used as antiresorptive drugs, leading to progressive destruction of jaw bones [1]. Bisphosphonates have been regarded as the drug of choice for the treatment of osteoporosis and are also used in the management of many other non-malignant and malignant conditions because of their effect on the inhibition of bone resorption by impeding the function of osteoclasts [2,3,4,5].

On the other hand, bisphosphonates have been reported to accumulate and affect the dissolution of bone mineral contents [1]. Moreover, bisphosphonates inhibit the turnover and blood supply in bones leading to osteonecrosis, including the jaw [6]. Symptoms of MRONJ are delayed healing after oral surgery, including tooth extraction, jaw osteonecrosis with incomplete healing of soft tissue with exposed bone, and risk of infection [1,2,3,5,6,7]. Moreover, when MRONJ involves the region beyond the alveolar bone, it may lead to mandibular fracture, extra oral fistula extending to the inferior border of the mandible or maxillary sinus and neuropathies [8,9]. These symptoms have significant adverse effects on the oral health quality of life of the patient [6,10,11,12,13,14]. Therefore, general dental practitioners should minimize the risk of development of MRONJ-like symptoms and be able to perform early diagnosis of MRONJ for prevention [15]. Physicians should also refer the patient to the dentist for proper oral examination and prophylactic dental treatment, when necessary, and be advised on maintaining good oral hygiene before zoledronic acid or denosumab therapy, as recommended by The European Society for Medical Oncology [16,17].

Although the mechanisms of MRONJ onset is still unclear, the following factors are suggested to be associated with or to contribute to increase the risks for MRONJ onset: suppression of osteoclast activity, inhibition of angiogenesis, reduced blood flow, decrease in immunity, increase in oral bacterial infection, and inhibition of epithelial cell migration [18,19,20].

Currently, a multidisciplinary approach (oncologist, maxillofacial surgeon, and dentist) is performed for the management of MRONJ depending on the degree of severity [21]. Treatment modalities vary from non-invasive procedures, such as patient education of oral hygiene, oral antibacterial mouth rinse, teriparatide (TPTD), a combination of pentoxifylline and tocopherol, to surgical interventions, such as buccal fat pad (BFP) flaps, hyperbaric oxygen therapy, low-intensity laser (LIL) treatment and platelet-rich plasma (PRP) therapy [19,22,23].

Promoting bone remodeling is one of the main strategies for the management of bone necrosis in MRONJ. Bone morphogenetic proteins (BMPs), which belong to the transforming growth factor (TGF)-β superfamily, are secreted cytokines that regulate the fate and function of different cell types and play a central role in organogenesis, including bone and cartilage development [24,25]. BMPs were identified as potent bone formation-inducing agents based on their ability to induce de novo bone formation in mice [19]. On the other hand, disruption of BMP signaling causes skeletal abnormalities and vascular disorders [19]. Moreover, BMP signaling plays a crucial role in the differentiation of mesenchymal stem cells (MSCs) into osteoblasts and bone formation. To date, at least 20 types of BMPs have been identified, and BMP-2 is one of the most promising molecules for bone regeneration [6]. Indeed, the U.S. Food and Drug Administration (FDA) has approved the mammalian cell-derived recombinant human BMP-2 (C-rhBMP-2, INFUSE, Medtronic Sofamor Danek, Memphis, TN, US) for the treatment of cervical spinal fusion in 2002 [26], and sinus augmentation and alveolar ridge augmentations associated with extraction sockets in 2007 [27,28,29]. Our research group has also succeeded in producing Good Manufacturing Practices (GMP)-grade rhBMP-2 using an *Escherichia coli* production system (E-rhBMP-2). Yano et al. have also revealed that the biological capacity of E-rhBMP-2 is almost equivalent to those of C-rhBMP-2 [30].

Beta-tricalcium phosphate (β-TCP), a synthetic bone graft substitute. is known to have osteoinductive, osteogenic abilities and cell-mediated resorption and has been used as a scaffold in the dental field for guided bone regeneration to prevent the ingrowth of soft tissues into bone defects [31,32]. Previously, Ono et al. [33] and Mikai et al. [34] demonstrated the advantages of hard biomaterial scaffolds (β-TCP) with BMP-2 for endogenous cell recruitment to induce bone formation around dental implants and in the tooth extraction socket in MRONJ-like mice models.

BMP-2 is well known for its ability to enhance both bone turnover in vivo by regulation of osteoblastic differentiation, and bone resorption by regulation of osteoclastic differentiation directly via BMP receptors [35,36]. Moreover, BMPs are important regulators of angiogenesis [37]. It is speculated that BMP-2 can prevent the onset of MRONJ and treat MRONJ-associated bone necrosis. Previously, we evaluated the effect of BMP-2 on the prevention and treatment of MRONJ, and demonstrated that local administration of BMP-2/β-TCP in the tooth extraction sockets significantly induced bone formation and reduced bone necrosis [26]. However, it is still unclear whether administration of cyclophosphamide (CY)/ zoledronate (ZA) can induce necrosis of the bone surrounding the tooth extraction sockets and whether local administration of BMP-2/β-TCP can improve the quality of the bone surrounding the tooth extraction sockets.

## 2. Results

### 2.1. Effect of CY/ZA Administration before Tooth Extraction

To understand whether CY/ZA administration could induce a MRONJ-like condition, the alveolar bone around the maxillary first molars were analyzed after three weeks of CY/ZA administration in mice (Figure 1A). The number of empty lacunae was counted in the surrounding alveolar bone at different distances (0–100 µm, 100–200 µm and 200–300 µm) from the lateral boundaries of the tooth root socket (Figure 1B). There was no significant difference in the number of empty lacunae in the control group (no therapy) compared to CY/ZA therapy group (Figure 1C,D). From these results, it was clarified that administration of CY/ZA for only three weeks cannot induce alveolar bone necrosis.

### 2.2. Effect of CY/ZA Administration 2 Weeks after Tooth Extraction

Next, to investigate the effects of both CY/ZA administration on bone necrosis after tooth extraction, histomorphometric analyses were performed at two weeks after extraction of the maxillary first molar (Figure 2A). Calculation of the number of empty lacunae revealed that the CY/ZA-administered group showed a significant increase in empty lacunae compared to the control group (Figure 2C,D). Additionally, there was impaired healing in the bone surrounding the tooth extraction socket. These results indicate that both administration of CY/ZA and surgery (tooth extraction) are necessary to induce osteonecrosis in the tissue surrounding the tooth extraction sockets. On the other hand, the control group showed normal bone healing in the extraction sockets and increased lacunae number in its surroundings.

### 2.3. Effect of BMP-2/β-TCP Transplantation on the Alveolar Bone Surrounding the Tooth Extraction Socket in the Prevention of the Onset of MRONJ-like Symptoms

To investigate whether BMP-2/β-TCP transplantation affects the bone surrounding the tooth extraction socket in the MRONJ-like prevention model, the BMP-2/β-TCP complex was transplanted into the tooth extraction socket immediately after tooth extraction, and analyses were performed after four weeks post-transplantation. CY/ZA were administered continuously until seven weeks, according to a previously described protocol [34] (Figure 3A). The quality of the bone surrounding the tooth extraction socket in the MRONJ-like prevention model was evaluated by the number of empty lacunae and the bone mineral density (BMD) of furcation and mesial areas adjacent to the tooth extraction socket. The BMD in BMP-2 group [furcation area (2.37 ± 0.06), mesial area (2.42 ± 0.07)] was significantly higher than in control group [furcation area (2.06 ± 0.09), mesial area (1.94 ± 0.10)] (Figure 3B,C). The numbers of empty lacunae were significantly decreased in the BMP-2/β-TCP group (117 ± 28.4, 61.6 ± 10.4) compared to those in the control group (209 ± 12.7, 114 ± 18.4, respectively at 0–100 µm and 100–200 µm from the surface of tooth extraction sockets). No significant difference in the number of empty lacunae was observed between the two groups at 200–300 µm (control: 115 ± 21.3 vs. BMP-2/β-TCP: 114 ± 22.1) (Figure 3F,G). Together, the results suggest that the transplantation of BMP-2/β-TCP enhances bone formation not only in the tooth extraction sockets [34], but also reduces bone necrosis around the tooth extraction sockets in the prevention of MRONJ symptoms in an MRONJ-like prevention models in mice.

### 2.4. Effect of BMP-2/β-TCP Transplantation on the Alveolar Bone Surrounding the Tooth Extraction Socket in the Treatment of MRONJ-like Symptoms

To explore the effect of BMP-2/β-TCP on MRONJ-associated bone loss around the tooth extraction socket after CY/ZA administration, BMP-2/β-TCP was transplanted after two weeks of extraction of the maxillary first molar, with a concomitant suspension of the CY/ZA administration (Figure 4A). Analysis was performed after four weeks of BMP-2/β-TCP transplantation (Figure 4A). In the micro-CT analysis, there was no significant difference in BMD between the BMP-2-transplanted group [furcation area (2.30 ± 0.14), mesial area (2.27 ± 0.13)] and the control group [furcation area (2.33 ± 0.15), mesial area (2.29 ± 0.14)] (Figure 4B,C). However, the number of empty lacunae in the BMP-2/β-TCP-transplanted group (138 ± 15.7, 100 ± 15.4, 93.1 ± 16.3, at 0 µm to 100 µm, 100 µm to 200 µm and 200 µm to 300 µm from the surface of tooth extraction socket, respectively) was significantly lower compared to that in the control group (233 ± 8.08, 144 ± 10.6, 195 ± 34.4, at 0 µm to 100 µm, 100 µm to 200 µm and 200 µm to 300 µm from the surface of tooth extraction socket, respectively) (Figure 4F,G). Note that the tooth extraction socket in the control group was filled with soft tissue, while BMP-2/β-TCP transplantation could promote partial bone repair. These results suggest that transplantation of BMP-2/β-TCP could ameliorate the bone necrosis around the extraction socket in the treatment of MRONJ-like symptoms.

## 3. Discussion

MRONJ is characterized by the necrosis of the maxillary or mandibular bone due to adverse drug reactions of antiresorptive and antiangiogenic therapies [17]. Moreover, previous clinical reports revealed that the onset of MRONJ symptoms is associated with invasive dental procedures, including tooth extraction [17,38]. Our data also demonstrated that the onset of MRONJ-associated symptoms was followed by tooth extraction in MRONJ model mice (Figure 1 and Figure 2). Moreover, local delivery of BMP-2 promoted not only new bone formation in the tooth extraction sockets [34], but also reduced bone necrotic symptoms in the bone surrounding the tooth extraction sockets in both MRONJ-like prevention and treatment models in mice (Figure 3 and Figure 4). On the other hand, it is still controversial whether BMP-2 could enhance BMD [39,40]. In this study, the BMD of the BMP-2-transplanted group was significantly higher than that in the control group in the MRONJ-like prevention model, but there was no significant difference in BMD between the two groups in the MRONJ-like treatment model. Our data revealed that BMP-2 has beneficial effects through the suppression of the osteonecrosis-like changes in the alveolar bone surrounding the tooth extraction socket and the inducing of new bone formation in tooth extraction sockets in MRONJ-like models regardless of the continuation or discontinuation of the administration of CY/ZA. In this study, a strict design of the control group was essential to minimize errors and obtain more valid results. An Escherichia coli-derived rhBMP-2 (E-rhBMP-2)/β-TCP is now being developed for clinical applications, analysis of its effect on the prevention or treatment for MRONJ is critically important to expand its clinical applications. 

Conservative surgical management, wound debridement and curettage have been performed for the treatment MRONJ in the clinical setting [41]. More recently, platelet-rich plasma (PRP) and platelet-rich fibrin (PRF), which are autogenous sources of many growth factors, including platelet-derived growth factor (PDGF), transforming growth factor-beta (TGF-β), epidermal growth factors (EGF) and vascular endothelial growth factors (VEGF), have often been used in the regeneration of extraction sockets [42]. For instance, Mauceri et al. has evaluated the effect of PRP on reducing the risk of MRONJ for patients receiving antiresorptive drugs, and concluded that PRP application may contribute to a reduction in the occurrence of MRONJ [43]. Kailas et al. reported that PRP enhanced the osteogenic response in initial bone healing, although there was no beneficial effect at the late wound healing period in extraction sockets [44]. They have also reported that PRP strongly improved the soft tissue healing in the extraction socket. Because BMP-2 could not enhance soft tissue healing, a combination therapy of PRP and BMP-2/β-TCP could be more effective than the single therapies. Despite these promising results, a recent systematic review could not yet find significant evidence supporting the application of platelet concentrates for the prevention and treatment of MRONJ [42]. 

Artificial bone graft materials, such as hydroxyapatite and β-TCP, have not been used for the repair of bone defects, mainly because the onset of MRONJ is considered to be related to enhanced suppression of bone remodeling [45], and these artificial materials could not activate bone remodeling. Therefore, in this study, no artificial bone graft material was transplanted in the control group.

One of the possible causes of MRONJ is the inhibition of bone remodeling with excessive suppression of osteoclast activity and the accompanying decrease in bone metabolic activity [45]. Promoting bone remodeling is, therefore, one of the main strategies for the management of bone necrosis in MRONJ. On the other hand, BMP-2 is known to not only activate the bone remodeling process but also stimulate the expression of the mineralization-related genes. There is increasing evidence of well-studied in vivo and in vitro reports of BMP-2 involvement in osteogenesis through the BMP-2 receptor in both osteoblasts and osteoclasts [35,36,46,47]. Owing to these evidence-based research findings, the clinical use of rhBMP-2, which works as a substitute for a bone graft, was approved by the FDA in 2002 [48]. As a consequence, clinical trials using rhBMP-2 have increased in both orthopaedic and dental surgical procedures [39,49,50], and the FDA has approved the utilization of BMP-2 in anterior lumbar interbody spinal fusions, cranioplasties and maxillofacial reconstructive surgery [51]. Our research group has also succeeded in developing the E-rhBMP-2 in collaboration with Osteopharma Inc. (Osaka, Japan) for large scale manufacturing, and is currently undergoing clinical trials for delivery of E-rhBMP-2 in the oral region. Based on the results of this study, we suppose that BMP-2 has therapeutic potential for the prevention and treatment of MRONJ-like symptoms.

Moreover, the development and implementation of novel biomaterials which create space-making are important to the delivery and release of BMP-2 to the reconstruction area. Previous studies used BMP-2/hydrogel [48] and BMP-2/collagen [52] to evaluate the effect of BMP-2 in the treatment of MRONJ-associated symptoms. However, the effectiveness of soft biomaterials was not satisfactory because of their limitation to create space for bone formation. Ono et al. [33] reported that E-rhBMP-2/β-TCP induced bone formation in a porcine maxillary sinus floor elevation model. Moreover, Mikai et al. [34] also revealed the effectiveness of E-rhBMP-2/β-TCP in inducing bone formation in tooth extraction sockets. Therefore, in this study, we also focused on the combined effect of BMP-2/β-TCP in the bone quality of the surrounding alveolar bone in MRONJ-like mouse models.

## 4. Materials and Methods

### 4.1. Preparation of E-rhBMP2/β-TCP Complex

E-rhBMP2/β-TCP complex was prepared by 1.5 mg of porous β-TCP (Superpore, particle size 0.6–1.0 mm, porosity 75%, HOYA, Tokyo, Japan) dissolved in 2.5 μL of 0.5 mM HCl containing E-rhBMP-2 (Osteopharma Inc., Osaka, Japan) and incubated for five minutes at room temperature. 

### 4.2. Animal Model

C57BL/6J mice (8-week-old to 12-week-old females) were obtained from CLEA Japan Inc. (Osaka, Japan). To create MRONJ-like lesions in mice mimicking the human condition [17], a combined administration of Cyclophosphamide (CY) and Zoledronate (ZA) was performed as described [53]. Briefly, ZA (0.05 mg/kg, Zometa, Novartis, Stein, Switzerland) and CY (150 mg/kg, C7397; Sigma-Aldrich, St. Louis, MO, USA) were injected, respectively, subcutaneously and intraperitoneally, twice a week for three weeks. Saline solution was also injected as a control.

Maxillary first molars were extracted under general anaesthesia with isoflurane (Pfizer, New York, NY, USA) at three weeks after CY/ZA administration. To create the MRONJ-like prevention model, E-rhBMP-2/β-TCP was transplanted into tooth extraction sockets immediately after tooth extraction and CY/ZA were continuously administered twice a week for seven weeks. 

In the MRONJ-like treatment model, CY/ZA were administered twice a week for two weeks after tooth extraction. The tooth extraction sockets were curetted by a dental probe and E-rhBMP-2/β-TCP was transplanted into the tooth extraction sockets. The CY/ZA administration was suspended/terminated after transplantation of E-rhBMP2/β-TCP. The control groups received no transplantation.

The animal experiment protocol used in this study (OKU-2019254, OKU-2020380) was approved by the Okayama University Research Committee. Throughout the experimental period, animal care and experimental protocols were performed under the guideline of the Okayama University Animal Research Committee.

### 4.3. Micro-CT Analysis

The samples were fixed with 4% paraformaldehyde (PFA; Merck, Kenilworth, NJ, USA) and scanned by micro-computed tomography (micro-CT, SkyScan 1174, Bruker, Kontich, Belgium) as described previously [34]. The scanning parameters were set to 6.5 μm voxel size, 50 kVp, 800 μA, 1 mm aluminum filter, angular rotation step 0.7°, 180° scanning, 261 projections, and an exposure time of 4 s with a total scan duration of 37 min. Transmission images were reconstructed using SkyScan NReconc software (Bruker). To measure the cross-sectional volumetric BMD, the region of interest (100 μm square area, ROI) adjacent to the furcation and mesial areas of the root socket of the first maxillary molar was measured volumetrically through 25 serial sections from the sagittal data set by SkyScan CTan software (Bruker).

### 4.4. Histological Analysis

Fixed samples were decalcified with Morse solution (FUJIFILM Wako Pure Chemical Corporation, Osaka, Japan) for one week and embedded in paraffin. Sections of 5 μm were stained with Hematoxylin and Eosin (HE staining: 1% eosin Y solution, Delafield Hematoxylin: Muto Kagaku Co., Ltd., Tokyo, Japan). All images were taken with a BZ-710 microscope (Keyence, Osaka, Japan). Quantitative analyses of the number of empty osteocyte lacunae in the bone area surrounding tooth extraction sockets (between 0 μm to 100 μm, 100 μm to 200 μm, and 200 μm to 300 μm from the surface of the tooth extraction sockets, respectively) were measured with BZ analyzer software (Keyence).

### 4.5. Statistical Analysis

The statistical significance of the data was compared by using the unpaired Student’s *t*-test. Graph Pad Prism v.9 (GraphPad Software, La Jolla, CA, USA) was used for the analyses. Significance levels were as follows: * *p* < 0.05, ** *p* < 0.01, *** *p* < 0.001, ns no significant difference.

## 5. Conclusions

In summary, according to previous findings of BMP-2 in the bone remodeling capacity and properties of β-TCP, we showed for the first time the effectiveness of local transplantation of BMP-2/β-TCP in inducing an increase in bone formation in tooth extraction sockets and improving the bone quality in the surrounding tissue in MRONJ-like prevention and treatment models in mice. Although large animal studies are needed for the confirmation of the clinical application of BMP-2/β-TCP in the management of MRONJ-like symptoms, BMP-2 could be considered as an important therapy for the management of MRONJ.

## Figures and Tables

**Figure 1 ijms-22-12823-f001:**
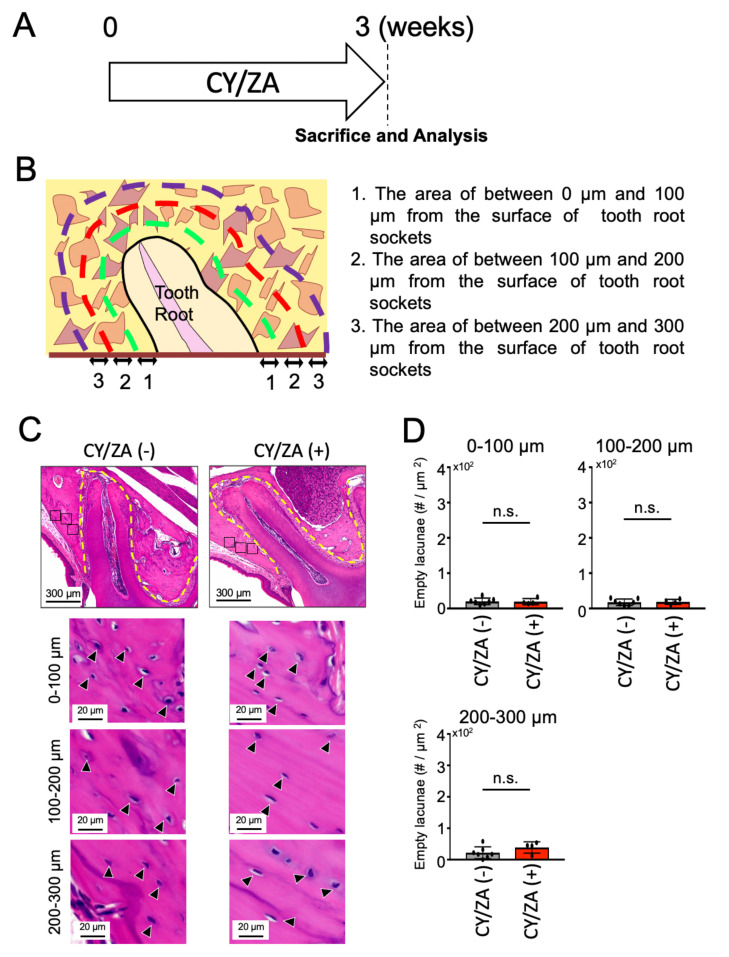
Effect of CY/ZA administration on the alveolar bone around the tooth root. (**A**) Experimental design; Administration of CY/ZA for three weeks before tooth extraction of the maxillary first molar. CY: cyclophosphamide, ZA: zoledronate. (**B**) Graphic image of the selected bone areas around the tooth root socket. Green, Red and purple lines represent the areas at 0 µm to 100 µm, 100 µm to 200 µm and 200 µm to 300 µm from the surface of the tooth socket, respectively. (**C**) Hematoxylin and Eosin (HE)-stained sagittal images of the bone surrounding the tooth sockets after CY/ZA administration. The control group was injected with saline. The yellow line indicates the tooth root socket. Black and white arrowheads represent lacunae with and without cells, respectively. (**D**) The graph shows the quantitative analysis of empty lacunae in the bone surrounding the tooth root socket. Bars represent the mean ± standard deviation. ns: no significant difference. unpaired Students’ *t*-test, *n* = 4–7.

**Figure 2 ijms-22-12823-f002:**
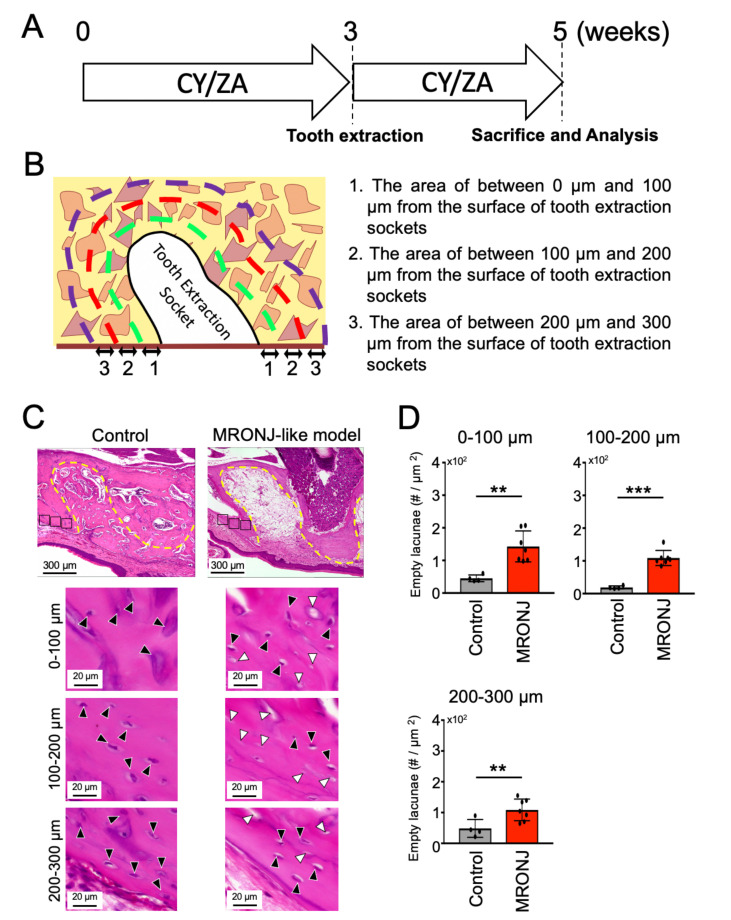
Effect of CY/ZA administration on the alveolar bone around the tooth extraction socket after two weeks of tooth extraction. (**A**) Experimental design: CY/ZA were administered for three weeks and continued for two weeks after extraction of the maxillary first molar. CY: cyclophosphamide, ZA: zoledronate. (**B**) Graphic image of the selected bone areas around the tooth extraction socket. Green, Red and purple lines represent the areas at 0 µm to 100 µm, 100 µm to 200 µm and 200 µm to 300 µm from the surface of the tooth socket, respectively. (**C**) HE-stained sagittal images of the bone surrounding the tooth extraction socket of control and MRONJ-like model (CY/ZA administration). Yellow line indicates the tooth extraction socket. Black and white arrowheads represent lacunae with and without cells in the bone, respectively. (**D**) The graph represents the quantitative analysis of empty lacunae in the bone surrounding the tooth extraction socket. The number of empty lacunae was significantly increased in the experimental group compared to that in the control group. The bars represent the mean ± standard deviation. *** *p* < 0.001, ** *p* < 0.01. unpaired Students’ *t*-test. *n* = 4–7.

**Figure 3 ijms-22-12823-f003:**
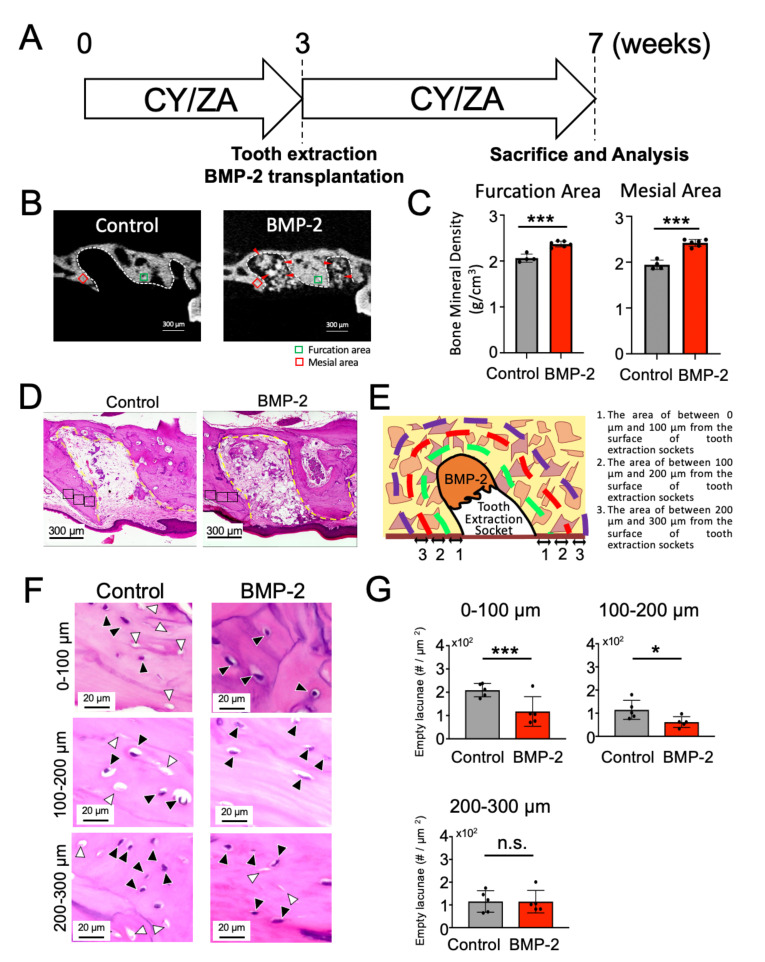
Effect of BMP-2/β-TCP transplantation on the bone surrounding the tooth extraction socket in a MRONJ-like prevention model. (**A**) Experimental design: BMP-2/β-TCP complex was transplanted into the socket after extraction of the maxillary first molar at three weeks. After four weeks post-transplantation, mice were sacrificed and submitted for analysis. CY/ZA was administered continuously throughout seven weeks. CY: cyclophosphamide, ZA: zoledronate. (**B**) Representative micro-CT images of the tooth extraction socket and the surrounding alveolar bone in the control and BMP-2/β-TCP-transplanted groups. Red arrow indicates residual BMP-2/β-TCP. White dotted line shows the tooth extraction socket. Green and red squares mark the furcation and mesial areas of the alveolar bone (100 µm^2^, ROI) adjacent to the tooth extraction socket, respectively. (**C**) Quantitative analysis of bone mineral density (BMD) of the mesial and furcation areas of the alveolar bone adjacent to the tooth extraction socket. The bar graph was presented as mean ± standard deviation; *** *p* < 0.001, unpaired Students’ *t*-test, *n =* 4–6. (**D**) HE-stained sagittal images of bone surrounding the tooth extraction sockets of control and BMP-2/β-TCP-transplanted group. Yellow line indicates the tooth extraction socket. (**E**) Graphic image of the selected bone area around the tooth extraction sockets. Green, red and purple lines represent the areas at 0 µm to 100 µm, 100 µm to 200 µm and 200 µm to 300 µm from the surface of tooth extraction socket, respectively. (**F**) HE-stained sagittal images of the selected bone area around the tooth extraction sockets at 0 µm to 100 µm, 100 µm to 200 µm and 200 µm to 300 µm from the surface of tooth extraction socket, respectively, in the control and BMP-2/β-TCP-transplanted group. Black and white arrowheads represent lacunae with and without cells, respectively. (**G**) The graph shows the quantitative analysis of empty lacunae in the bone surrounding the tooth extraction socket. The number of empty lacunae was significantly decreased in the BMP-2/β-TCP-transplanted group compared to that in the control group. The bar represents the mean ± standard deviation. *** *p* < 0.001, * *p* < 0.01, ns: no significant difference. unpaired Students’ *t*-test, *n =* 5.

**Figure 4 ijms-22-12823-f004:**
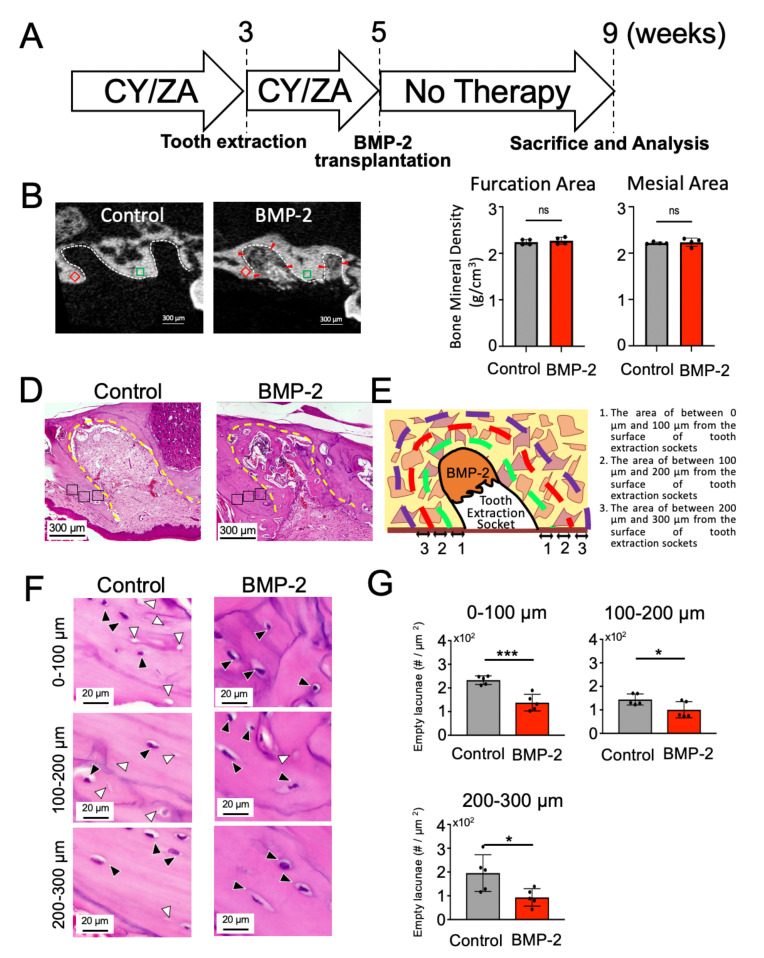
Effect of BMP-2/β-TCP on the bone surrounding the tooth extraction socket in the MRONJ-like treatment model. (**A**) Experimental design: Extraction of the maxillary first molar of mice was performed after three weeks of CY/ZA administration. Two weeks after extraction, BMP2/β-TCP was transplanted into the extraction socket and administration of CY/ZA was terminated. CY: cyclophosphamide, ZA: zoledronate. (**B**) Representative micro-CT images of the surrounding alveolar bone and tooth extraction socket in the control and BMP-2/β-TCP-transplanted groups. Red arrowheads indicate residual BMP-2. White dotted line shows the tooth extraction socket. Green and red square marked the furcation and the mesial area of the alveolar bone (100 μm^2^, ROI) adjacent to the tooth extraction socket, respectively. (**C**) Quantitative analysis of bone mineral density (BMD) of the mesial and the furcation area of the alveolar bone adjacent to the tooth extraction socket. The bars in the graph represent the mean ± standard deviation. ns- no significant. unpaired Students’ *t*-test, *n =* 4–6. (**D**) HE-stained sagittal images of the bone surrounding the tooth extraction socket of control (No drugs) and BMP-2/β-TCP-transplanted groups. Yellow line indicates the tooth root socket. (**E**) Graphic image of selection of bone area around the tooth extraction sockets. Green, red and purple lines represent the areas at 0 µm to 100 µm, 100 µm to 200 µm and 200 µm to 300 µm from the surface of tooth extraction socket, respectively. (F) HE-stained sagittal images of the selected bone area around the tooth extraction sockets at 0 µm to 100 µm, 100 µm to 200 µm and 200 µm to 300 µm from the surface of tooth extraction socket, respectively, in the control and BMP-2/β-TCP-transplanted group. Black and white arrowheads represent lacunae with and without cells, respectively. (**G**) The graph indicates the quantitative analysis of empty lacunae in the bone surrounding the tooth extraction socket. The number of empty lacunae was significantly decreased in the BMP-2/β-TCP-transplanted group compared to that in the control group. The bars represent the mean ± standard deviation. *** *p* < 0.001, * *p* < 0.05, unpaired Students’ *t*-test, *n =* 5.

## Data Availability

Data may be available upon request to the corresponding author.

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
