# Peer review of "Suppression of Bone Necrosis around Tooth Extraction Socket in a MRONJ-like Mouse Model by E-rhBMP-2 Containing Artificial Bone Graft Administration"

_ijms, 2021, doi:10.3390/ijms222312823_

Round 1

Reviewer 1 Report

The manuscript submitted to IJMS entitled “Suppression of bone necrosis around tooth extraction socket in a MRONJ-like mouse model by E-rhBMP-2 containing artificial bone graft administration” is an original research article which aim to examine the therapeutic potential of BMP-2 in combination with the hard osteoinductive biomaterial, β-tricalcium phosphate (β-TCP), in the prevention and treatment of alveolar bone loss around tooth extraction sockets in Medication-related osteonecrosis of the jaw (MRONJ)-like mice models.

On my opinion the article is interesting, well written, with good English. Anyway, some improvements are needed.

  • English language: Minor spell check is required.
  • Abstract: Please structure abstract to attract readers’ attention.
  • Introduction: I would suggest improving this section inserting a sentence on clinical sign related to MRONJ:<< Major and minor clinical signs in patients treated with drugs related to osteonecrosis of the jaw, such as neuropathy, are important for the prevention, diagnosis and management of MRONJ [https://doi.org/1016/j.jormas.2018.04.006].>>
  • Results: This section has been properly prepared.
  • Discussion: Please discuss the use of platelet concentrates in the prevention of osteonecrosis of the jaws [https://doi.org/10.1016/j.jcms.2020.01.014].
  • Materials and Methods: This section has been properly prepared.

Please insert a “Conclusion” section.

After making the indicated changes, the article will be suitable for publication.

Thanks for the opportunity to review this manuscript.

Author Response

Reviewer 1

Comment 1

English language: Minor spell check is required.

Response 1: The authors thank the detailed comment addressed by the reviewer. The entire text has been edited for English spelling and grammar, and for clarity.

Comment 2  

Abstract: Please structure abstract to attract readers’ attention.

Response 2: The authors thank the comment addressed by the reviewer. The abstract has been structured, according to the reviewer’s suggestion and the IJMS guidelines.

Comment 3

Introduction: I would suggest improving this section inserting a sentence on clinical sign related to MRONJ:<< Major and minor clinical signs in patients treated with drugs related to osteonecrosis of the jaw, such as neuropathy, are important for the prevention, diagnosis and management of MRONJ [https://doi.org/1016/j.jormas.2018.04.006].>>

Response 3: The authors thank the important comment addressed by the reviewer. Based on this comment, the authors have added the following sentence describing other clinical signs of MRONJ, including neuropathies (page 2, lines: 58-60):

“Moreover, when MRONJ involves the region beyond the alveolar bone, it may lead to mandibular fracture, extra oral fistula extending to the inferior border of mandible or maxillary sinus and neuropathies [8, 9].”

Additional references:

Haviv, Y.; Geller, Z.; Mazor, S.; Sharav, Y.; Keshet, N.; Zadik, Y., Pain characteristics in medication-related osteonecrosis of the jaws. Support Care Cancer 2021, 29, (2), 1073-1080.

Ruggiero, S. L., Diagnosis and Staging of Medication-Related Osteonecrosis of the Jaw. Oral Maxillofac Surg Clin North Am 2015, 27, (4), 479-87.

Nevertheless, the reference (Miyamoto et al., 2021) suggested by the reviewer was related to osteoradionecrosis of the jaw and not MRONJ. Therefore, to avoid confusion to the readers, the authors decided to not include the suggested reference. We hope that the reviewer could understand and accept the authors’ decision.

Comment 4

Discussion: Please discuss the use of platelet concentrates in the prevention of osteonecrosis of the jaws [https://doi.org/10.1016/j.jcms.2020.01.014].

Response 4: The authors thank the important comment addressed by the reviewer, and have added the following sentence regarding the use of platelet concentrates in the prevetion and treatment of MRONJ (page: 11, lines: 276-290). The suggested reference was also included in the manuscript.

“More recently, platelet-rich plasma (PRP) and platelet-rich fibrin (PRF), which are autogenous sources of many growth factors, including platelet-derived growth factor (PDGF), transforming growth factor-beta (TGF-b), epidermal growth factors (EGF) and vascular endothelial growth factors (VEGF), have often used in the regeneration of extraction sockets [42]. For instance, Mauceri et al. has evaluated the effect of PRP on reducing the risk of MRONJ for patients receiving antiresorptive drugs and concluded that PRP application may contribute to reduce the occurrence of MRONJ [43]. Kailas et al. reported that PRP enhanced the osteogenic response in initial bone healing although there was no beneficial effect at late wound healing period in extraction sockets [44]. They have also reported that PRP strongly improved the soft tissue healing in extraction socket. Because BMP-2 could not enhance soft tissue healing, a combination therapy of PRP and BMP-2/b-TCP could be more effective than the single therapies. Despite these promising results, a recent systematic review could not yet find a significant evidence supporting the application of platelet concentrates for the prevention and treatment of MRONJ [42].”

Comment 5

Please insert a “Conclusion” section      

Response 5: The authors thank the comment addressed by the reviewer, and have inserted a Conclusion section at the end of the manuscript (page: 13, lines: 381-389).

Reviewer 2 Report

Dear Authors

  • in introduction there should be some information on how important it is to do the checkup (dental) before using bioposphonates
  • materials and method's session should be 2nd one (not 4th)
  • It would be very interesting to add within the discussion some details about PRP after tooth extraction with use of bioposphates  (eg. Pietruszka P, ChruÅ›cicka I, DuÅ›-Ilnicka I, Paradowska-Stolarz A. PRP and PRF-Subgroups and Divisions When Used in Dentistry. J Pers Med. 2021;11(10):944. Published 2021 Sep 23. doi:10.3390/jpm11100944  and  Mauceri, R.; Panzarella, V.; Pizzo, G.; Oteri, G.; Cervino, G.; Mazzola, G.; Di Fede, O.; Campisi, G. Platelet-Rich Plasma (PRP) in Dental Extraction of Patients at Risk of Bisphosphonate-Related Osteonecrosis of the Jaws: A Two-Year Longitudinal Study. Appl. Sci. 202010, 4487. https://doi.org/10.3390/app10134487), as 
    PRP and PRF are the newest trends in dental treatment.
  • Diagrams D in the figures 1 and 2 could be a little bit larger
  • 25 out of 43 references are from the last 10 years, which is not bad, but could be improved.

Overall, the paper is interesting and in my opinion deserves publishing.

Author Response

Reviewer 2

Comment 1

in introduction there should be some information on how important it is to do the checkup (dental) before using bioposphonates

Response 1: The authors thank the important comment made by the reviewer, and have added the following sentence in the Introduction section (page: 2, lines: 62-67).

  “Therefore, general dental practitioners should minimize the risk of development of MRONJ-like symptoms and be able to perform early diagnosis of MRONJ for prevention [15]. Physicians also should refer the patient to the dentist for proper oral examination and prophylactic dental treatment, when necessary, and be advised on maintaining good oral hygiene before zoledronic acid or denosumab therapy, as recommended by The European Society for Medical Oncology [16, 17].”

References:

Nicolatou-Galitis, O., Schiødt, M., Mendes, R. A., Ripamonti, C., Hope, S., Drudge-Coates, L., Niepel, D., & Van den Wyngaert, T. (2019). Medication-related osteonecrosis of the jaw: definition and best practice for prevention, diagnosis, and treatment.Oral surgery, oral medicine, oral pathology and oral radiology,127(2), 117–135. https://doi.org/10.1016/j.oooo.2018.09.008

Coleman, R., Body, J. J., Aapro, M., Hadji, P., Herrstedt, J., & ESMO Guidelines Working Group (2014). Bone health in cancer patients: ESMO Clinical Practice Guidelines.Annals of oncology : official journal of the European Society for Medical Oncology,25 Suppl 3, iii124–iii137. https://doi.org/10.1093/annonc/mdu103

  1. Ruggiero, S. L., Dodson, T. B., Fantasia, J., Goodday, R., Aghaloo, T., Mehrotra, B., O'Ryan, F., & American Association of Oral and Maxillofacial Surgeons (2014). American Association of Oral and Maxillofacial Surgeons position paper on medication-related osteonecrosis of the jaw--2014 update.Journal of oral and maxillofacial surgery : official journal of the American Association of Oral and Maxillofacial Surgeons,72(10), 1938–1956. https://doi.org/10.1016/j.joms.2014.04.031

Comment 2

materials and method's session should be 2nd one (not 4th)

Response 2:  The authors thank the important comment made by the reviewer. However, as per the guidelines for manuscript preparation, IJMS instruct the authors to place the “Materials and Methods section” in the 4th place, after the Discussion section.

IJMS Instruction for authors: https://www.mdpi.com/journal/ijms/instructions#preparation

Comment 3

It would be very interesting to add within the discussion some details about PRP after tooth extraction with use of bioposphates  (eg. Pietruszka P, ChruÅ›cicka I, DuÅ›-Ilnicka I, Paradowska-Stolarz A. PRP and PRF-Subgroups and Divisions When Used in Dentistry. J Pers Med. 2021;11(10):944. Published 2021 Sep 23. doi:10.3390/jpm11100944  and  Mauceri, R.; Panzarella, V.; Pizzo, G.; Oteri, G.; Cervino, G.; Mazzola, G.; Di Fede, O.; Campisi, G. Platelet-Rich Plasma (PRP) in Dental Extraction of Patients at Risk of Bisphosphonate-Related Osteonecrosis of the Jaws: A Two-Year Longitudinal Study. Appl. Sci. 2020, 10, 4487. https://doi.org/10.3390/app10134487), as 
PRP and PRF are the newest trends in dental treatment.

Response 3: The authors thank the important comment made by the reviewer, and have added the following paragraph in the Discussion section  (page: 11, lines: 276-290).

“More recently, platelet-rich plasma (PRP) and platelet-rich fibrin (PRF), which are autogenous sources of many growth factors, including platelet-derived growth factor (PDGF), transforming growth factor-beta (TGF-b), epidermal growth factors (EGF) and vascular endothelial growth factors (VEGF), have often used in the regeneration of extraction sockets [42]. For instance, Mauceri et al. has evaluated the effect of PRP on reducing the risk of MRONJ for patients receiving antiresorptive drugs and concluded that PRP application may contribute to reduce the occurrence of MRONJ [43]. Kailas et al. reported that PRP enhanced the osteogenic response in initial bone healing although there was no beneficial effect at late wound healing period in extraction sockets [44]. They have also reported that PRP strongly improved the soft tissue healing in extraction socket. Because BMP-2 could not enhance soft tissue healing, a combination therapy of PRP and BMP-2/b-TCP could be more effective than the single therapies. Despite these promising results, a recent systematic review could not yet find a significant evidence supporting the application of platelet concentrates for the prevention and treatment of MRONJ [42].”

Regarding the references suggested by the reviewer, the second reference (Mauceri et al.) is strongly related to the main message of this paper, and has been added in the revised manuscript. The first reference, however, is a short review of the methods to obtain PRP and PRF, and therefore, is not directly related to our paper. The authors opted to not include the first reference in the revised manuscript. We hope that the reviewer could understand and accept our decision.

Comment 4

Diagrams D in the figures 1 and 2 could be a little bit larger

Response 4: The authors thank the detailed comments addressed by the reviewer, and have enlarged the graphs in Fig.1D and Fig.2D.

Comment 5

25 out of 43 references are from the last 10 years, which is not bad, but could be improved.

Response 5: The authors thank the detailed comment made by the reviewer, and have updated the references. Below are some of the uptaded references that were added or used to replace old ones.

He, L., Sun, X., Liu, Z., Qiu, Y., & Niu, Y. (2020). Pathogenesis and multidisciplinary management of medication-related osteonecrosis of the jaw. International journal of oral science12(1), 30. https://doi.org/10.1038/s41368-020-00093-2

Graham R., Russell G. Bisphosphonates: The first 40 years. Bone. 49, 1: 2-19 (2011)

On, S. W., Cho, S. W., Byun, S. H., & Yang, B. E. (2021). Various Therapeutic Methods for the Treatment of Medication-Related Osteonecrosis of the Jaw (MRONJ) and Their Limitations: A Narrative Review on New Molecular and Cellular Therapeutic Approaches. Antioxidants (Basel, Switzerland)10(5), 680. https://doi.org/10.3390/antiox10050680

Lorenzo-Pouso, A. I., Bagán, J., Bagán, L., Gándara-Vila, P., Chamorro-Petronacci, C. M., Castelo-Baz, P., Blanco-Carrión, A., Blanco-Fernández, M. Á., Álvarez-Calderón, Ó., Carballo, J., & Pérez-Sayáns, M. (2021). Medication-Related Osteonecrosis of the Jaw: A Critical Narrative Review. Journal of clinical medicine10(19), 4367. https://doi.org/10.3390/jcm10194367